# Effect of COVID-19 on Falls in a Residential Care Facility for the Elderly: Longitudinal Observational Study

**DOI:** 10.3390/jcm14176229

**Published:** 2025-09-03

**Authors:** Ana González-Castro, Raquel Leirós-Rodríguez, Marta Nistal-Martínez, Ernesto Bodero-Vidal, José Alberto Benítez-Andrades, Pablo Hernandez-Lucas

**Affiliations:** 1Nursing and Physical Therapy Department, University of León, Astorga Ave., 24401 Ponferrada, Spain; angonc@unileon.es; 2SALBIS Research Group, Nursing and Physical Therapy Department, University of León, Astorga Ave., 24401 Ponferrada, Spain; 3SACYL Castilla y León Health Services, Pseo. de Zorrilla 1, 47007 Valladolid, Spain; marta.nistal@dipuleon.es (M.N.-M.); ebv75y@ad.sms.carm.es (E.B.-V.); 4ALBA Research Group, Department of Electric, Systems and Automatics Engineering, University of Leon, Campus of Vegazana s/n, 24071 León, Spain; jbena@unileon.es; 5Faculty of Physiotherapy, University of Vigo, Campus A Xunqueira, 36005 Pontevedra, Spain; phernandez@uvigo.es

**Keywords:** coronavirus infections, accidental falls, accidents, housing for the elderly, frail elderly

## Abstract

**Background/Objectives:** During the Coronavirus Disease 2019 (COVID-19) pandemic, various safety measures were implemented in elderly care facilities in Spain. These measures led to a reduction in physical activity and increased supervision of residents, often resulting in the suspension of outings from the facility. The objective of this study was to assess the influence of COVID-19 preventive measures on the number and characteristics of falls among elderly individuals living in a residential care facility in Spain. **Methods:** A retrospective longitudinal observational study was conducted from 2018 to 2021. Over these four years, data related to falls were collected from a residential care facility for the elderly. Both patient characteristics and fall characteristics were recorded. **Results:** The average age of the 48 residents continuously institutionalized between 2018 and 2021 was 85.8 ± 5.1 years. A total of 364 falls occurred during the study period, with 68% of them taking place in 2019 and 2020. Although the number of falls increased during the COVID-19 pandemic, the characteristics of the falls did not change. However, residents who experienced falls were increasingly accompanied at the time of the event. **Conclusions:** Based on the data collected from the elderly care facility analyzed in this study, falls increased during the COVID-19 pandemic, but the measures implemented in residential care facilities do not appear to have altered the characteristics of the falls, except for the greater presence of companionship.

## 1. Introduction

In April 2021, the World Health Organization (WHO) reported alarming figures regarding falls and their consequences; approximately 37.3 million falls worldwide in that year were severe enough to require medical attention [1]. Moreover, these falls result in 684,000 deaths annually, making them the second leading cause of death from unintentional injuries [1,2]. The WHO also reported that individuals aged 60 years and older are most affected by fatal falls. Even when such incidents are non-fatal, they frequently lead to increased morbidity and a higher likelihood of institutionalization [1,3]. Other sources confirm that one in three people over the age of 65 experiences at least one fall per year, with the incidence rising significantly among those over 80 years old [2,4].

Falls represent a major global public health concern [5,6]. From a socioeconomic perspective, the annual cost of falls and their consequences in the United States is estimated at USD 50 billion [3,7]. In Spain, according to the National Statistics Institute, there were 3707 deaths due to unintentional fall-related injuries in 2022 [8].

Among older adults, falls that might otherwise be considered “minor” can pose a serious threat to independence, quality of life, and even survival [3,9]. In approximately 30% of cases, falls in this population result in bruises and/or fractures, leading to functional decline, dependence, fear of falling again, depression, and reduced physical activity levels [8,10,11].

Specifically, in residential care facilities, half of the residents experience at least one fall annually [12]. Furthermore, more than half of these incidents result in injury [13]. The high prevalence of falls in such settings is partly attributable to risky behaviors among residents, including rushing, carelessness, wearing inappropriate footwear, and physical inactivity, all of which contribute to deteriorating general health [14]. In addition to risky behaviors, falls among institutionalized individuals are associated with frailty, dependence, and cognitive impairment. Conversely, institutionalized older adults who retain greater autonomy in ambulation, personal hygiene, and medication management have a lower risk of fall and fall-related injuries [15].

On 21 March 2020, the Spanish government issued COVID-19 prevention measures for elderly care facilities and socio-health centers in response to the public health crisis caused by the pandemic [16]. These measures addressed both healthcare and non-healthcare staff within the facilities, protocols for the placement and isolation of COVID-19 patients, cleaning and disinfection procedures, coordination for diagnosis and monitoring, and clinical guidelines for healthcare professionals regarding patient care [16]. These measures led to the isolation of a large proportion of residents, disrupting their daily routines and activities and altering the care they received.

The analysis of fall risk in institutionalized older adults requires a multifactorial approach that integrates biological, psychological, social, and environmental variables [17]. The biopsychosocial model recognizes that the likelihood of falling is influenced not only by intrinsic factors, such as physical frailty, balance impairments, and comorbidities, but also by psychosocial elements, including social isolation and disruptions to daily routines, as well as the physical and organizational characteristics of residential facilities [18]. The COVID-19 pandemic simultaneously altered these components by limiting mobility, modifying routines, affecting supervision, and increasing loneliness [19,20]. This convergence highlights the importance of adopting such a framework to better understand and interpret the observed findings.

Given the vulnerability of institutionalized older adults and the profound changes induced by pandemic-related measures, investigating how these factors influence fall patterns is essential to guide targeted prevention strategies and strengthen preparedness for future public health emergencies.

For these reasons, this study was undertaken to analyze the prevalence and characteristics of falls in a residential care facility for older adults during the COVID-19 pandemic in comparison with previous years.

## 2. Materials and Methods

### 2.1. Study Design

A retrospective longitudinal observational study was conducted from 2018 to 2021 in a residential care facility for older adults in Castilla y León, Spain. Over this four-year period, data on falls were collected at the Santa Luisa Public Care Center for the Elderly in León. Sampling was performed using a census approach, including all residents who met the eligibility criteria during the study period, in order to maximize internal validity and minimize potential selection bias.

The study protocol was approved by the University of León Research Ethics Committee (code: ULE-031-2021). All participants provided written informed consent in accordance with the Declaration of Helsinki (2013 version). Clinical trial registration was not applicable.

### 2.2. Procedure and Study Variables

The data analyzed were obtained from the medical records of the facility’s residents. Within the Rehabilitation Service, information on the characteristics of falls experienced by residents has been systematically recorded since 2018.

Inclusion criteria comprised residents who remained continuously institutionalized throughout the entire four-year study period. Exclusion criteria included residents who were not continuously institutionalized during the study period, those with incomplete or inconsistent fall records, and residents with temporary stays for rehabilitation or prolonged hospital admissions that interrupted data collection.

Data collection followed a standardized protocol and was carried out by trained Rehabilitation Service staff, who systematically reviewed medical records and supplemented the information with internal incident reports to ensure completeness and accuracy.

A manual record was maintained that included patient characteristics, such as age, sex, year of admission, and number of falls experienced. It also documented fall-related details, including the presence or absence of a third person at the time of the fall (defined as the presence of a staff member or an external visitor; other residents were not included in this category), the location within the facility where the fall occurred, the time of occurrence, the fall mechanism, contributing factors, the activity being performed, any assistive devices used, and injuries sustained.

According to institutional records, nighttime staffing levels and supervision protocols remained unchanged between the pre-pandemic and pandemic periods.

At the end of the data collection period, the information was analyzed to compare results across years. This enabled the creation of a dataset to assess the influence of COVID-19-related measures implemented in residential care facilities on fall risk and fall characteristics.

### 2.3. Statistical Analysis

The choice of statistical tests was determined by the non-normal distribution of the data and the small sample size. Descriptive statistics included the mean, median, and interquartile range. The Kolmogorov–Smirnov test was applied to assess the normality of residuals, and Levene’s test was used to verify the homogeneity of variances. The Mann–Whitney U test was employed for outcome analysis. To compare the number of falls per subject between the periods 2018–2019 and 2020–2021, the non-parametric Wilcoxon signed-rank test for related samples was used, given that the data were not normally distributed and corresponded to paired measurements.

Potential confounding factors were reviewed descriptively by examining patterns over time and relevant contextual variables; however, no multivariate adjustment was performed due to the study’s scope and small sample size.

A significance threshold of *p* < 0.05 was established. All analyses were conducted using Stata version 16.0 for MacOS^®^ (StataCorp LLC, College Station, TX, USA).

## 3. Results

The sample comprised 48 older residents who were continuously institutionalized from 2018 to 2021 (Table 1). The mean age of the participants was 85.8 ± 5.1 years.

Over the four-year period, a total of 364 falls were recorded, with 68% occurring in 2019 and 2020 (Figure 1). Specifically, the total number of falls increased progressively by 44% from 2018 to 2019 and by 9% from 2019 to 2020. Conversely, falls decreased by 60% from 2020 to 2021. Statistically significant differences were found in the number of falls between the periods 2018–2019 and 2020–2021 (Z = 2.10, *p* = 0.036).

Regarding the timing of falls, the highest proportion occurred either during the night or in the hours adjacent to it (between 8:00 and 10:00 a.m. or 7:00 and 10:00 p.m.). This pattern remained consistent throughout the four years analyzed (Table 2).

Regarding the location of falls, residents’ rooms were the most common site over the four-year period, followed by bathrooms. Areas like the dining room, stairs, and hallways recorded significantly fewer falls.

The characteristics of falls related to modifiable external factors remained unchanged throughout the study period. Across all four years, more than 60% of falls were not associated with external factors; however, wet floors accounted for 5–11% of total falls.

Another variable analyzed was the activity being performed at the time of the fall. Among the categories considered (walking, sit-to-stand transfer, personal hygiene, or falling from bed), most falls occurred while residents were walking. Regarding the presumed cause of the fall, several possibilities were assessed (involvement of a third person, dizziness, instability, or tripping/slipping), with instability being the most frequently reported cause. Again, no differences were observed between years (Table 2).

The type of assistance used at the time of the fall was also evaluated, including cane, walker, or wheelchair use. Most falls occurred when residents were not using any walking aid.

The fall mechanism did not vary over the study period. Residents fell forward, backward, sideways, or while seated, regardless of the year.

Finally, the variable of accompaniment (whether the resident was accompanied at the time of the fall) was examined. The results showed a progressive increase in accompaniment from 2018 to 2021, with 2020 registering the highest proportion of accompanied falls (Figure 2).

## 4. Discussion

The objective of this study was to analyze the prevalence and characteristics of falls in a residential care facility for older adults during the COVID-19 pandemic. Although the number of falls progressively increased over the four-year period analyzed, their characteristics remained largely unchanged during the pandemic period.

The only variable that showed a significant change over time was “accompaniment.” The findings indicate that residents who experienced falls were frequently accompanied at the time of the event, with this prevalence increasing progressively. This trend paralleled the overall rise in fall incidence and the proportion of falls occurring in the presence of another person.

An important consideration regarding the variable of accompaniment is that it referred exclusively to the presence of staff members or external visitors and not other residents. The present findings highlight that falls may occur even when residents are accompanied, suggesting that supervision alone does not necessarily guarantee prevention. In a qualitative study, long-term care staff noted that “even with all the measures we put in place to prevent falls, they still fall by the time we get to them,” reflecting that some incidents may occur regardless of supervision efforts [21].

Previous research has emphasized the importance of supervision in reducing the risk of adverse events in nursing homes [22,23]. However, our results suggest that the quality and type of accompaniment may be more relevant than its mere presence. This underscores the need for future studies to explore in greater depth how different forms of accompaniment (e.g., healthcare staff, auxiliary staff, family members) influence fall risk and whether specific strategies can enhance its protective effect.

The increase in accompaniment despite the isolation measures imposed during the COVID-19 period may be explained by various factors. From a social perspective, the measures implemented in elderly care facilities raised concerns about their potential impact on the quality of care and services provided to residents [24]. However, the present findings suggest that despite the isolation of a substantial proportion of residents, their care and accompaniment were not negatively affected.

One possible explanation for the parallel increase in falls and accompaniment lies in the psychosocial factors affecting residents. During the COVID-19 pandemic, several studies investigated its impact on psychosocial risks in elderly care facilities [25,26]. These studies concluded that both staff and residents were adversely affected by a lack of information and by fear of infection and even death [27]. Such psychosocial risks may have contributed to the increase in falls by influencing the physical and emotional state of residents through heightened fear and anxiety [28].

Our findings are consistent with recent international evidence describing changes in the temporal patterns of falls during pandemics or periods of strict mobility restrictions [29,30]. A meta-analysis published in 2023 on the incidence of falls in nursing homes during COVID-19 restrictions confirmed a higher occurrence of nocturnal falls, attributed to reduced daytime mobility and altered nighttime supervision routines [31]. Importantly, in the facility where this study was conducted, nighttime staffing levels and supervision protocols remained unchanged between the pre-pandemic and pandemic periods. Therefore, the increase in nocturnal falls cannot be attributed to reduced supervision. Alternative explanations may involve factors not directly measured in this study, such as sleep disturbances and disrupted routines, which have been associated with an increased risk of falls in older adults [32].

Recommended preventive strategies include optimizing nighttime supervision, implementing adaptive lighting to improve visibility during nocturnal movements, and reviewing environmental risk factors in private rooms [31,33].

Regarding the characteristics of falls, and contrary to the findings of Vlaeyen et al. [34], this study found that most falls occurred during the night. In contrast, Vlaeyen’s observational study reported that 62% of recorded falls took place during the morning and afternoon [34]. These findings were later corroborated in 2020 by Molés et al. [35], who reported that most falls occurred either forward or backward. In the present study, however, no significant differences were observed in the mechanisms of falls among residents. Finally, with respect to the potential influence of external factors, and similar to the research conducted by Carballo et al. [36], no clear association was found between external factors and the occurrence of falls in residents.

In relation to the mechanisms of falls, our results are consistent with recent large-scale reviews indicating that the direction of a fall remains stable across different contexts, although the severity of consequences may vary depending on the use of assistive devices and the presence of environmental hazards [10,37]. The most recent international guidelines recommend multifactorial prevention programs that combine strength and balance training with regular medication review [10,38].

There is currently a substantial body of research on fall risk and its predisposing factors [6,39,40]. The vast majority of these studies conclude that both balance and motor control are critical elements in preventing, experiencing, or recovering from fall [33]. In 2024, González et al. [41] reinforced this view through a systematic review that included 22 interventions aimed at developing predictive models for fall risk. In creating these models, in addition to employing innovative methods, such as artificial intelligence, they analyzed various accelerometric variables that reflect participants’ balance and stability [41].

Accordingly, reduction in physical activity during the isolation period is associated with a deterioration in overall health. This decline may be accompanied by impairments in fundamental abilities, such as motor control, balance, and stability, which could explain the progressive increase in falls observed during the COVID-19 period. Given that these are decisive factors in fall risk, they may account for the rise in falls recorded during the pandemic [42,43].

Likewise, inactivity and reduced physical stimulation during confinement periods have been associated with accelerated frailty progression and an increased risk of falls [44]. Recent systematic reviews underscore the importance of integrating both physical and psychosocial interventions, recommending a combination of supervised exercise programs, medication review, and targeted environmental modifications [45,46].

The findings of this study underscore the need to implement multifactorial strategies to prevent falls in residential care facilities. In particular, preventive programs should include structured physical activity even during confinement periods, enhance nighttime supervision through adaptive lighting solutions, and systematically assess environmental risk factors in private rooms. Furthermore, fall prevention protocols should incorporate psychosocial support for both residents and caregivers, given the impact of isolation and disrupted routines observed during the pandemic. From a policy perspective, these measures highlight the importance of embedding fall prevention strategies into contingency plans for future public health emergencies.

Finally, it is important to acknowledge the potential influence of confounding variables, such as seasonal variations in fall incidence, undocumented changes in internal facility policies, and unmeasured environmental factors, which may have influenced the observed results [47,48]. Considering these factors in future research will help strengthen causal interpretations and guide the development of more precise preventive strategies.

In addition to these considerations, this study has several limitations that should be acknowledged. First, the sample size was small and drawn from a single residential care facility, which limits the generalizability of the results to other populations or settings. Second, information on falls was obtained from the facility’s internal records, which may involve reporting bias, particularly for events not directly observed by staff. Furthermore, potential confounding variables, such as seasonal variations, undocumented changes in facility policies, or unmeasured environmental factors, were not systematically controlled for and may have influenced the incidence and characteristics of falls. These limitations underscore the need to interpret the results with caution and to design future multicenter studies with larger samples and more comprehensive control of contextual and environmental factors.

## 5. Conclusions

This study found that the number of falls increased during the COVID-19 period, although their characteristics did not differ from those observed in the years prior to the pandemic. In other words, the measures implemented in residential care facilities during the COVID-19 lockdown did not substantially alter fall characteristics.

However, the observed increase in nighttime falls is consistent with recent systematic reviews and meta-analyses reporting changes in fall timing during mobility restrictions, highlighting the influence of circadian factors and care routines on fall risk.

These findings support the need to implement targeted preventive measures in residential care facilities, such as optimizing nighttime supervision, installing adaptive lighting systems, reviewing environmental risk factors, and integrating multifactorial programs that combine strength and balance training with regular medication review.

From a policy perspective, the results point to the importance of developing structured physical activity programs that can be maintained during confinement periods, implementing fall prevention protocols tailored to high-risk residents, and integrating mental health support strategies for both residents and caregivers. From a public health standpoint, these measures should be embedded into contingency and preparedness plans for future health emergencies, ensuring a multidimensional approach that addresses the physical, environmental, and psychosocial determinants of fall risk.

Moreover, given that the association between increased accompaniment and the psychological burden of isolation remains tentative, future research should incorporate validated psychometric tools to assess this relationship more accurately and examine its impact on fall risk. Further studies with larger sample sizes are also needed to allow for more reliable extrapolation of the findings and to analyze the variable of accompaniment in greater detail, particularly in relation to measures implemented during the COVID-19 period.

## Figures and Tables

**Figure 1 jcm-14-06229-f001:**
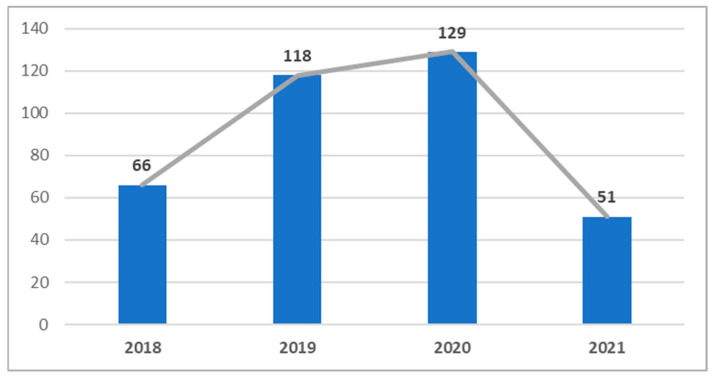
Number of falls per year.

**Figure 2 jcm-14-06229-f002:**
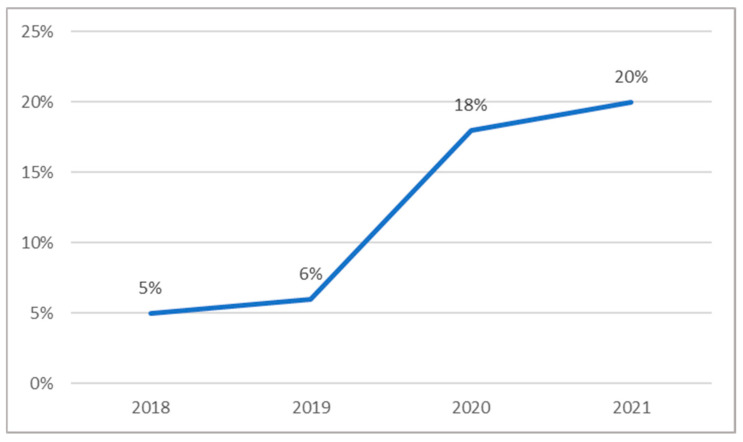
Proportion of falls that occurred while older adults were accompanied by someone else.

**Table 1 jcm-14-06229-t001:** Characteristics of continuously institutionalized residents from 2018 to 2021.

Variable	Mean (*n* = 48)	Median	Interquartile Range
Age	85.8	87	10
Number of years in nursing home	8.7	7	7
Total number of falls during time spent in nursing home	11.3	6	11
Average total number of falls during time spent in nursing home	1.7	0.7	1.8
Number of falls in 2018–2019	3.9	1	3
Number of falls in 2020–2021	5	2	6

**Table 2 jcm-14-06229-t002:** Characteristics of falls.

Variable	2018	2019	2020	2021	All
Number of falls	66	118	129	51	364
Accompaniment [n (percentage)]					
Yes	3 (4.5%)	7 (5.9%)	23 (17.8%)	10 (19.6%)	43 (11.8%)
No	63 (95.5%)	111 (94.1%)	102 (79.1%)	41 (80.4%)	317 (87.1%)
Unknown	0 (0)	0 (0)	4 (3.1%)	0 (0)	4 (1.1%)
Time of the fall [n (percentage)]					
8.00–9.59	5 (5.6%)	15 (12.7%)	18 (14%)	9 (17.6%)	47 (12.9%)
10.00–12.59	3 (4.5%)	24 (36.4%)	26 (20.2%)	9 (17.6%)	62 (17%)
13.00–14.59	4 (6.1%)	10 (8.5%)	14 (10.9%)	7 (13.7%)	35 (9.6%)
15.00–16.59	6 (9.1%)	9 (7.6%)	13 (10.1%)	6 (11.8%)	34 (9.3%)
17.00–18.59	1 (1.5%)	10 (8.5%)	11 (8.5%)	3 (5.9%)	25 (6.9%)
19.00–21.59	12 (18.2%)	14 (11.9%)	21 (16.3%)	11 (21.6%)	58 (16%)
22.00–7.59	20 (30.3%)	36 (30.5%)	26 (20.2%)	6 (11.8%)	88 (24.2%)
Unknown	15 (22.7%)	0 (0)	0 (0)	0 (0)	15 (4.1%)
Location of the fall [n (percentage)]					
Room	30 (43.5%)	66 (55.9%)	59 (45.7%)	22 (43.1%)	177 (48.6%)
Hallway	5 (7.6%)	28 (23.7%)	30 (23.3%)	11 (21.6%)	74 (20.3%)
Bathroom	14 (21.2%)	12 (10.2%)	16 (12.4%)	5 (9.8%)	47 (12.9%)
Dining room	4 (6.1%)	2 (1.7%)	6 (4.7%)	1 2%)	13 (3.6%)
Lounge	5 (7.6%)	7 (5.9%)	12 (9.3%)	8 (15.7%)	32 (8.8%)
Chapel	0 (0)	1 (0.8%)	0 (0)	0 (0)	1 (0.3%)
Stairs	1(1.5%)	0 (0)	1 (0.8%)	0 (0)	2 (0.6%)
Gym	0 (0)	0 (0)	0 (0)	0 (0)	0 (0)
Gardens	2 (3%)	2 (1.7%)	0 (0)	3 (5.9%)	7 (1.9%)
Cafeteria	0 (0)	0 (0)	0 (0)	1 (2%)	1 (0.3%)
Physical therapy room	0 (0)	0 (0)	5 (3.9%)	0 (0)	5 (1.4%)
Unknown	5 (7.6%)	0 (0)	0 (0)	0 (0)	5 (1.4%)
Cause of the fall [n (percentage)]					
Instability	28 (42.4%)	70 (59.3%)	65 (50.4%)	25 (49%)	188 (51.6%)
Trip or slip	23 (34.8%)	24 (20.3%)	17 (13.2%)	8 (15.7%)	72 (19.8%)
Dizziness	4 (6.1%)	4 (3.4%)	3 (2.3%)	4 (7.8%)	15 (4.1%)
Third person influence	1 (1.5%)	1 (0.8%)	0 (0)	0 (0)	2 (0.6%)
Unknown	10 (15.2%)	19 (16.1%)	44 (34.1%)	14 (27.5%)	87 (23.9%)
Fall mechanism [n (percentage)]					
Forward	17 (25.8%)	22 (18.6%)	21 (16.3%)	10 (19.6%)	70 (19.2%)
Backward	12 (18.2%)	36 (30.5%)	23 (17.8%)	13 25.5%)	84 (23.1%)
Sideways	17 (25.8%)	27 (22.9%)	21 (16.3%)	12 (23.5%)	77 (21.2%)
Seated	13 (19.7%)	7 (5.9%)	21 (16.3%)	10 (19.6%)	51 (14%)
Unknown	7 (10.6%)	26 (22%)	43 (33.3%)	6 (11.8%)	82 (22.5%)
Activity during the fall [n (percentage)]					
Walking	36 (54.5%)	62 (52.5%)	52 (40.3%)	26 (51%)	176 (48.4%)
Sitting–standing	12 (18.2%)	19 (16.1%)	32 (24.8%)	14 (27.5%)	77 (21.2%)
Falling from bed	2 (3%)	10 (8.5%)	7 (5.4%)	3 (5.9%)	22 (6%)
Personal hygiene	1 (1.5%)	4 (3.4%)	1 (0.8%)	2 (3.9%)	8 (2.2%)
Other	11 (16.7%)	19 (16.1%)	16 (12.4%)	4 (7.8%)	50 (13.7%)
Rest	4 (6.1%)	4 (3.4%)	21 (16.3%)	2 (3.9%)	31 (8.5%)
Orthopedic elements during falls [n (percentage)]					
Cane	12 (18.2%)	7 (5.9%)	14 (10.9%)	9 (17.6%)	42 (11.5%)
Supervision	0 (0)	2 (1.7%)	4 (3.1%)	1 (2%)	7 (1.9%)
Crutches	0 (0)	0 (0)	0 (0)	0 (0)	0 (0)
Walker	13 (19.7%)	14 (11.9%)	45 (34.9%)	18 (35.3%)	90 (24.7%)
Wheelchair	3 (4.5%)	1 (0.8%)	4 (3.1%)	0 (0)	8 (2.2%)
None	32 (48.5%)	90 (76.3%)	35 (27.1%)	19 (17.6%)	176 (48.4%)
Unknown	6 (9.1%)	4 (3.4%)	27 (20.9%)	4 (7.8%)	41 (11.3%)
Influence of external factors [n (percentage)]					
Wet floors	7 (10.6%)	9 (7.6%)	7 (5.4%)	3 (5.9%)	26 (7.1%)
Objects on the floor	1 (1.5%)	7 (5.9%)	0 (0)	1 (2%)	9 (2.5%)
Unstable furniture	0 (0)	4 (3.4%)	2 (1.6%)	2 (3.9%)	8 (2.2%)
Lack of lighting	3 (4.5%)	4 (3.4%)	1 (0.8%)	1 (2%)	9 (2.5%)
Poorly used orthopedic aids	8 (12.1%)	4 (3.4%)	10 (7.8%)	3 (5.9%)	25 (6.9%)
Inadequate footwear	9 (13.6%)	13 (11%)	10 (7.8%)	1 (2%)	33 (9.1%)
Other	6 (9.1%)	13 (11%)	11 (8.5%)	7 (13.7%)	37 (10.2%)
No influence	32 (48.5%)	64 (54.2%)	88 (68.2%)	33 (64.7%)	217 (59.6%)

## Data Availability

The datasets used and/or analyzed during the current study are available from the corresponding author upon reasonable request.

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
