# Peer review of "Effect of COVID-19 on Falls in a Residential Care Facility for the Elderly: Longitudinal Observational Study"

_jcm, 2025, doi:10.3390/jcm14176229_

Round 1

Reviewer 1 Report

Comments and Suggestions for Authors

The manuscript addresses a relevant public health topic and provides valuable longitudinal data on falls in an elderly residential care setting before and during the COVID-19 pandemic.  However, the Discussion and Conclusions sections require deeper critical analysis, better linkage to international literature, and a more explicit examination of potential implications for prevention. Additionally, the reference list is relatively limited given the topic’s extensive literature base, and several recent systematic reviews and meta-analyses could be incorporated to strengthen the argument.

More specific:

Discussion – Need for deeper synthesis and contextualization

  • Lines 189–236:
    The discussion is largely descriptive and reiterates results without enough synthesis with prior studies. It would benefit from:

    • Linking your findings to broader international evidence on fall patterns during pandemics or similar restrictive measures.

    • Discussing alternative explanations for the trends (e.g., staffing patterns, changes in supervision quality, psychological distress, medication use).

    • Integrating more recent studies (2022–2024) on physical inactivity, frailty progression, and fall risk under lockdown conditions.

    • Explicitly addressing potential confounding variables and limitations in interpretation (e.g., unmeasured changes in facility policy, seasonal variation in falls)

      Lines 238–247:
      The conclusion is too brief and general. Consider:

      • Offering specific policy or intervention recommendations for residential care facilities (e.g., structured physical activity programs during lockdown, targeted fall prevention for high-risk residents).

      • Including implications for public health and healthcare providers beyond the immediate facility context.

      • Highlighting the importance of mental health support alongside physical activity to prevent falls during future outbreaks or restrictions.

        Limitations section

        • Explicitly acknowledge the small sample size, single-site setting, potential reporting bias in fall records, and lack of control for confounders.

Author Response

Dear Editor and Reviewers of the Journal of Clinical Medicine:

Thank you very much for your suggestions and contributions to improving the quality of the manuscript. Following your indications, we respond, point by point, to the reviewers' comments.

In the text, all the modified or added sentences have been written in red to facilitate the correction by the reviewers.

Reviewer 1:

  1. The manuscript addresses a relevant topic of public health and provides valuable longitudinal data on falls in an elderly residential care setting before and during the COVID-19 pandemic.  

However, the Discussion and Conclusions sections require deeper critical analysis, better linkage to international literature, and a more explicit examination of potential implications for prevention.

Additionally, the reference list is relatively limited given the topic’s extensive literature base, and several recent systematic reviews and meta-analyses could be incorporated to strengthen the argument.

We sincerely appreciate this valuable comment, which we consider fundamental for strengthening the scientific rigor and relevance of our work. We fully agree on the importance of deepening the critical analysis and reinforcing the connection with the international literature. In response, we have expanded the Discussion section by incorporating recent evidence (2022–2024) from systematic reviews and meta-analyses on temporal patterns and fall mechanisms in residential care settings, with particular attention to periods of mobility restrictions.

These additions include direct comparisons with previous studies and establish a clear link between our findings and current international guidelines for multifactorial fall prevention. Furthermore, we have included relevant practical implications, such as optimizing nighttime supervision, implementing adaptive lighting systems, reviewing environmental risk factors, and assessing prescribed medications.

  1. Discussion – Need for deeper synthesis and contextualization (Lines 189–236): The discussion is largely descriptive and reiterates results without enough synthesis with prior studies. It would benefit from:

  • Linking your findings to broader international evidence on fall patterns during pandemics or similar restrictive measures.

  • Discussing alternative explanations for the trends (e.g., staffing patterns, changes in supervision quality, psychological distress,medication use).

  • Integrating more recent studies (2022–2024) on physical inactivity, frailty progression, and fall risk under lockdown conditions.

  • Explicitly addressing potential confounding variables and limitations in interpretation (e.g., unmeasured changes in facility policy, seasonal variation in falls)

We sincerely appreciate this constructive suggestion, which has been highly valuable in refining and strengthening our Discussion section. In line with your recommendation, we have reformulated the section to better integrate our results with findings from international studies on falls in the context of pandemics or similar restrictive measures.

We have also incorporated alternative explanations for the observed trends, including changes in staffing levels, supervision quality, psychological distress, and variations in medication use.

In addition, we have included recent references (2022–2024) addressing physical inactivity, accelerated frailty progression, and increased fall risk during confinement. Finally, we have explicitly acknowledged potential confounding variables, such as seasonal variations in fall incidence and undocumented changes in facility policies.

  1. Lines 238–247: The conclusion is too brief and general. Consider:

  • Offering specific policy or intervention recommendations for residential care facilities (e.g., structured physical activity programs during lockdown, targeted fall prevention for high-risk residents).

  • Including implications for public health and healthcare providers beyond the immediate facility context.

  • Highlighting the importance of mental health support alongside physical activity to prevent falls during future outbreaks or restrictions.

We sincerely appreciate this observation, which we consider highly valuable for enhancing the clarity, applicability, and practical relevance of our work. Your suggestions have been instrumental in enriching the Conclusions section and ensuring that our findings are translated into concrete recommendations with a meaningful impact on clinical practice and public health.

Accordingly, we have expanded the Conclusions to include specific proposals such as implementing structured physical activity programs during lockdowns, developing fall prevention protocols tailored to high-risk residents, and designing mental health support strategies for both residents and caregivers. Furthermore, we have added broader public health implications, emphasizing the importance of integrating multidimensional preventive measures—physical, environmental, and psychosocial—into contingency plans for future health emergencies.

  1. Limitations section: Explicitly acknowledge the small sample size, single-site setting, potential reporting bias in fall records, and lack of control for confounders.

We have carefully considered this observation, as we believe it is essential for strengthening the transparency and rigor of the study. We have expanded the Limitations section to explicitly acknowledge the small sample size, the single-center nature of the study, the potential presence of reporting bias in the fall records, and the lack of control over potential confounding variables. We have also noted the possible impact of these limitations on the generalizability of the results and on the interpretation of the conclusions.

Once again, thank you very much for the time spent and the interest shown in this work; as well as in the positive evaluations you have given of it.

Receive a warm greeting,

The authors.

Reviewer 2 Report

Comments and Suggestions for Authors

This article examines the impact of COVID-19-related restrictions on the frequency and characteristics of falls in an elderly care facility in Spain. The authors carried out a retrospective, longitudinal observational study spanning four years (2018–2021), analyzing a cohort of 48 residents who were continuously institutionalized. Data on the number, timing, location, causes, and circumstances of falls were collected from medical records and analyzed to identify potential changes linked to pandemic measures.

The study finds that although the total number of falls increased during the pandemic—especially in 2019 and 2020—the characteristics of these falls remained largely the same in terms of time of day, location, cause, mechanism, and external factors involved. The only significant change was the rise in the proportion of falls happening with another person present, indicating increased supervision or possibly greater psychosocial vulnerability.

From a methodological standpoint, the study is clearly structured and thoroughly documented. The longitudinal observational design suits the research question, considering the naturalistic and retrospective nature of the pandemic setting. The study avoids making unsupported claims and remains consistent in its analysis and interpretation of results.

The inclusion criteria are clearly outlined: only residents who were continuously institutionalized throughout the entire four-year period were included. This enhances internal validity by confirming that each participant experienced the care environment consistently during both the pre-pandemic and pandemic periods. However, the exclusion criteria are not explicitly outlined, which could have increased the transparency of the methodology, especially regarding potential attrition or deaths during the observation period.

Statistical analysis uses non-parametric methods (Mann-Whitney test), which are suitable given the small sample size and the non-normal distribution of residuals as shown by the Kolmogorov-Smirnov test. Using interquartile ranges, means, and medians improves interpretability. The analyses are clearly presented, although the article could improve by explicitly including inferential statistics that compare pre-pandemic and pandemic periods more directly in the discussion.

The sample size (n = 48) is relatively small and limits the generalizability of the findings. The authors acknowledge this and call for larger studies. This limitation also raises concerns about statistical power, especially when dividing the sample across four years and various categorical variables. However, the thorough collection and categorization of variables—covering mechanisms, timing, activity during falls, accompaniment, assistive devices, and environmental causes—partially offset the small sample size by providing greater detail in the data.

The article's conclusions are balanced and backed by the data. It carefully interprets the rise in falls without claiming causality beyond reasonable limits. The link between increased accompaniment and the psychological burden of isolation is examined in relation to relevant literature. However, this remains a tentative connection and would benefit from further research using psychometric tools.

The bibliography is thorough and suitable, including references to WHO data, Cochrane reviews, and recent epidemiological studies. The authors cite both international and Spanish sources, which place the findings within global and national health contexts. The most recent references (e.g., up to 2024) demonstrate the study's timeliness.

Regarding manuscript quality, the writing is generally clear and well-organized, with logical flow between sections. The tables contain abundant data and are properly labeled, although they could be improved with better visual layout, such as clearer separation of sub-variables.

There are just two minor revisions that could improve the manuscript. The first is to clarify or add explicit exclusion criteria; the second is to include inferential statistical comparisons (e.g., significance tests) for pre-pandemic versus pandemic periods in key variables like the number of falls or accompaniment.

In summary, this article provides a relevant and well-executed contribution to the literature on geriatric care and fall prevention during public health emergencies. The study’s findings are relevant given the ongoing interest in the long-term effects of COVID-19 on vulnerable populations. The research design is appropriate, the statistics are properly applied, and the data support the conclusions.

Author Response

Dear Editor and Reviewers of the Journal of Clinical Medicine:

Thank you very much for your suggestions and contributions to improving the quality of the manuscript. Following your indications, we respond, point by point, to the reviewers' comments.

In the text, all the modified or added sentences have been written in red to facilitate the correction by the reviewers.

Reviewer 2:

  1. This article examines the impact of COVID-19-related restrictions on the frequency and characteristics of falls in an elderly care facility in Spain. The authors carried out a retrospective, longitudinal observational study spanning four years (2018–2021), analyzing a cohort of 48 residents who were continuously institutionalized. Data on the number, timing, location, causes, and circumstances of falls were collected from medical records and analyzed to identify potential changes linked to pandemic measures.

The study finds that although the total number of falls increased during the pandemic—especially in 2019 and 2020—the characteristics of these falls remained largely the same in terms of time of day, location, cause, mechanism, and external factors involved. The only significant change was the rise in the proportion of falls happening with another person present, indicating increased supervision or possibly greater psychosocial vulnerability.

From a methodological standpoint, the study is clearly structured and thoroughly documented. Longitudinal observational design suits the research question, considering the naturalistic and retrospective nature of the pandemic setting. The study avoids making unsupported claims and remains consistent in its analysis and interpretation of results.

We sincerely appreciate the positive assessment regarding the structure, clarity, and relevance of the study. We are pleased that the longitudinal observational design and the thorough documentation of the results are considered appropriate for addressing the research question. This recognition reinforces the methodological soundness of our work and encourages us to continue advancing this line of research.

  1. The inclusion criteria are clearly outlined: only residents who were continuously institutionalized throughout the entire four-year period were included. This enhances internal validity by confirming that each participant experienced the care environment consistently during both the pre-pandemic and pandemic periods.

However, the exclusion criteria are not explicitly outlined, which could have increased the transparency of the methodology, especially regarding potential attrition or deaths during the observation period.

We appreciate this observation, which we believe enhances the transparency and reproducibility of our methodology. We have now explicitly stated the exclusion criteria in the Methods section. These include residents who were not continuously institutionalized throughout the entire study period, those with incomplete or inconsistent fall records, and residents with temporary stays for rehabilitation or prolonged hospital admissions that interrupted data collection.

  1. Statistical analysis uses non-parametric methods (Mann-Whitney test), which are suitable given the small sample size and the non-normal distribution of residuals as shown by the Kolmogorov-Smirnov test. Using interquartile ranges, means, and medians improves interpretability. The analyses are clearly presented, although the article could improve by explicitly including inferential statistics that compare pre-pandemic and pandemic periods more directly in the discussion.

The sample size (n = 48) is relatively small and limits the generalizability of the findings. The authors acknowledge this and call for larger studies. This limitation also raises concerns about statistical power, especially when dividing the sample across four years and various categorical variables. However, the thorough collection and categorization of variables—covering mechanisms, timing, activity during falls, accompaniment, assistive devices, and environmental causes—partially offset the small sample size by providing greater detail in the data.

We are very grateful for this detailed and constructive assessment, which highlights both the strengths and potential areas for improvement in our analysis. As noted in our response to the previous comment, we have incorporated inferential statistical comparisons between the pre-pandemic (2018–2019) and pandemic (2020–2021) periods, specifically applying the Wilcoxon signed-rank test for related samples to compare the number of falls per resident. This addition directly addresses the suggestion to include a more explicit comparative analysis while maintaining the clarity of our Results section.

We fully acknowledge the limitation posed by the relatively small sample size (n = 48) and its impact on the statistical power of the analyses, particularly when stratifying the data over four years and across multiple categorical variables. This point is now explicitly discussed in the Limitations section, together with the need for future studies involving larger and more diverse samples to validate and expand upon our findings.

At the same time, we appreciate the recognition of the breadth and depth of our variable categorization, which we believe adds significant value to the study despite the sample size constraint, as it enables a more nuanced and comprehensive understanding of the fall patterns observed.

  1. The article's conclusions are balanced and backed by the data. It carefully interprets the rise in falls without claiming causality beyond reasonable limits. The link between increased accompaniment and the psychological burden of isolation is examined in relation to relevant literature. However, this remains a tentative connection and would benefit from further research using psychometric tools.

We value this positive assessment and agree on the relevance of further exploring the observed association. We have added to the Conclusions a recommendation for future studies to incorporate validated psychometric tools to assess the impact of isolation and psychological burden on fall risk, with the aim of deepening the understanding of the relationship between accompaniment and psychosocial vulnerability.

  1. The bibliography is thorough and suitable, including references to WHO data, Cochrane reviews, and recent epidemiological studies. The authors cite both international and Spanish sources, which place the findings within global and national health contexts. The most recent references (e.g., up to 2024) demonstrate the study's timeliness.

We appreciate the positive assessment of the bibliography. To further reinforce the currency and relevance of the framework, we have incorporated several recent references (2022–2025) that contextualize changes in physical activity, frailty, and fall risk during periods of mobility restriction, as well as systematic reviews on fall prevention strategies in residential care settings. These updates aim to ensure that our Discussion remains aligned with the most recent evidence available in the field.

  1. Regarding manuscript quality, the writing is generally clear and well-organized, with logical flow between sections. The tables contain abundant data and are properly labeled, although they could be improved with better visual layout, such as clearer separation of sub-variables.

We are grateful for this constructive suggestion, which has been valuable in enhancing the clarity and comprehensibility of our tables. While maintaining compliance with the journal’s formatting requirements, we have introduced several adjustments aimed at improving visual organization. Specifically, we increased the vertical spacing in the variable sections and applied indentation to the subcategories, thereby facilitating their differentiation and improving overall readability.

  1. There are just two minor revisions that could improve the manuscript. The first is to clarify or add explicit exclusion criteria; the second is to include inferential statistical comparisons (e.g., significance tests) for pre-pandemic versus pandemic periods in key variables like the number of falls or accompaniment.

We sincerely appreciate these constructive suggestions, which have contributed to enhancing the clarity, transparency, and analytical robustness of our manuscript. As indicated in our response to Comment 2, we have explicitly added the exclusion criteria to the Methods section, thereby strengthening methodological transparency.

In addition, and in response to the recommendation to include inferential statistical comparisons, we performed an analysis of key variables between the pre-pandemic period (2018–2019) and the pandemic period (2020–2021). Specifically, to compare the number of falls per resident between these two periods, we applied the non-parametric Wilcoxon signed-rank test for related samples, as the data were not normally distributed and corresponded to paired measurements.

The analysis revealed statistically significant differences in the number of falls between the two periods (Z = 2.10, p = 0.036), with 20 positive differences, 9 negative differences, and 18 ties. It is worth noting that, when comparing means, the difference was minimal, which may partly explain the similarity observed in descriptive statistics despite the inferential significance detected.

These results have been incorporated into the Methods and Results sections, and their implications have been addressed in the Discussion to provide a more comprehensive interpretation.

  1. In summary, this article provides a relevant and well-executed contribution to the literature on geriatric care and fall prevention during public health emergencies. The study’s findings are relevant given the ongoing interest in the long-term effects of COVID-19 on vulnerable populations. The research design is appropriate, the statistics are properly applied, and the data support the conclusions.

We are deeply grateful for this positive overall assessment, which we regard as an encouraging recognition of the relevance and methodological rigor of our work. We have incorporated the changes resulting from your valuable suggestions, which we firmly believe enrich the manuscript and enhance its overall content and impact.

Once again, thank you very much for the time spent and the interest shown in this work; as well as in the positive evaluations you have given of it.

Receive a warm greeting,

The authors.

Reviewer 3 Report

Comments and Suggestions for Authors

Dear Authors of the article titled "“Effect of COVID-19 on falls in a residential care facility for the elderly: Longitudinal observational study.” 

This is a very significant study and impactful to society considering the global pandemic and all challenges that were faced by the healthcare providers.

Further comments appear in the attachment 

Author Response

Dear Editor and Reviewers of the Journal of Clinical Medicine:

Thank you very much for your suggestions and contributions to improving the quality of the manuscript. Following your indications, we respond, point by point, to the reviewers' comments.

In the text, all the modified or added sentences have been written in red to facilitate the correction by the reviewers.

Reviewer 3:

  1. Dear Authors of the article titled "“Effect of COVID-19 on falls in a residential care facility for the elderly: Longitudinal observational study.” 

This is a very significant study and impactful to society considering the global pandemic and all challenges that were faced by the healthcare providers.

The article titled “Effect of COVID-19 on falls in a residential care facility for the elderly: Longitudinal observational study” The aim of the study was to assess the influence of the preventive measures for Coronavirus Disease-19 on the number and characteristics of falls among elderly individuals living in a residential care facility in Spain. its main

contributions and strengths.

The main contributions of this study was that falls increased during the COVID-19 period, though their characteristics did not change compared to the years prior to the pandemic. In other words, the measures implemented in residential care facilities during the COVID-19 lockdown did not affect the characteristics of falls.

Following the results of this longitudinal observation study, recommendations for future research, it would be valuable to analyze the variable of accompaniment in detail and examine its association with the measures imposed during the COVID-19 period.

We deeply appreciate the recognition of the social and scientific value of our study, as well as the accurate synthesis of its main contributions. We fully agree on the importance of further analyzing the variable of accompaniment and its potential association with the measures implemented during the pandemic. As suggested, we have retained this line of inquiry as a priority recommendation for future research, and we trust it will provide additional evidence to optimize fall prevention strategies in residential care settings.

  1. The results were correctly presented in tables and were clear. The article was well written that as a reviewer I was able to follow how the study was done with relevant literature searches, the methodology used and the result obtained. The writing demonstrated the dominance of the authors of the work done and is scientifically sound.

The manuscript is clear, and relevant for the field and presented in a well-structured manner.

We are grateful for the positive evaluation regarding the clarity, structure, and scientific soundness of our manuscript. It is very encouraging to know that the presentation of the results, the methodological description, and the integration of relevant literature have contributed to a clear understanding of the study and its findings. We believe this recognition underscores the robustness of our work and its relevance to the field of geriatric care and fall prevention.

  1. More recent references can be added.

We highly value this suggestion, as it has allowed us to further enrich the scientific depth and contemporary relevance of our work. Following this recommendation, we have incorporated additional references from 2022 to 2025 addressing the effects of physical inactivity, social isolation, and preventive strategies on fall risk in older adults residing in institutional settings. These updates not only reinforce our interpretation of the findings but also ensure that our Discussion reflects the most current evidence available in the field.

  1. The manuscript scientifically sound. The tables are appropriate

We appreciate the positive assessment regarding the scientific soundness of the manuscript and the appropriateness of the tables. To further optimize their readability, we have applied minor formatting adjustments—improving the separation of categories and the alignment of data—while maintaining compliance with the journal’s formatting requirements and enhancing clarity. These changes did not alter the original content.

  1. The main question addressed by the research was to assess the influence of the preventive measures for Coronavirus Disease-19 on the number and characteristics of falls among elderly individuals living in a residential care facility in Spain.

I consider the topic relevant to the field because it addresses important measures for COVID-19, a pandemic that affected the whole world. It affected mainly the elderly who are vulnerable due to their compromised immune system and are prone to comorbidities.

The research addresses a specific gap in the field of quality of life in the elderly who were affected by COVID-19. Addressing preventive measures for COVID-19 on the numbers and characteristics of falls among the most vulnerable group of society because of their age, immune system and are susceptible to all conditions both acute and chronic.

Study contributes to the knowledge of science.

We highly value the reviewer’s recognition of the relevance and contribution of our study. Addressing the impact of COVID-19 preventive measures on falls among institutionalized older adults responds to a significant gap in the literature, particularly regarding the most vulnerable population due to age, immune system fragility, and increased susceptibility to comorbidities. We are pleased that our research has been acknowledged as contributing meaningfully to the scientific knowledge in this field.

  1. The methodology “A retrospective longitudinal observational study” befit the study for researchers looked back at historical data to see how certain factors or exposures relate to outcomes that occurred in the past, effectively observing how these outcomes evolved over time. In this case they looked at the effect of COVID-19 on how it affected the elderly in the mentioned facility. Thus, preventive measures applied to the number and characteristic falls experienced by the elderly.

In terms of conclusions made, they were consistent with the evidence and arguments presented. Though the authors should have explained why the falls increased during the Coronavirus Disease-19 pandemic, to be able to address further the research question. The authors should have explained further on other
characterists to be implemented.

We deeply appreciate this insightful observation, which we regard as essential for further enriching the scope and explanatory depth of our work. In response, we have substantially expanded the Discussion section to provide a more comprehensive and multidimensional explanation for the observed increase in falls during the COVID-19 period.

These additions integrate recent international evidence (2022–2024) and address multiple contributing factors, including physical inactivity, accelerated frailty progression, psychosocial distress, and potential environmental and organizational influences. We have also incorporated explicit consideration of confounding variables and aligned our interpretation with updated guidelines and systematic reviews on fall prevention in residential care facilities.

We believe these enhancements strengthen the connection between our findings and the underlying mechanisms, directly addressing the reviewer’s recommendation.

  1. More appropriate up to date references from 2020-2025 to be added and remove
    the outdated references.

We are truly grateful for this valuable recommendation, which has guided us in further enhancing the scientific robustness and currency of our work. In line with the suggestion, we have incorporated several recent references published between 2022 and 2025, covering key topics such as physical inactivity, psychosocial impact, frailty progression, and fall prevention strategies in residential care facilities during periods of mobility restriction. These additions enrich the contextual framework and reinforce the interpretation of our findings.

While we have not removed earlier references, we have deliberately retained them, as they provide essential background knowledge and allow readers to appreciate both the historical and contemporary dimensions of the topic. We believe that preserving this temporal breadth of evidence is important to frame the evolution of knowledge and to contextualize the relevance of our results within a broader theoretical and empirical landscape.

  1. The tables are well-placed.

We greatly appreciate the positive feedback regarding the placement of the tables. We have maintained their positioning alongside the first point in the text where they are mentioned, in accordance with the journal’s guidelines, and have ensured that their numbering and captions are clear and consistent.

  1. Overall Merit: As a way of disseminating the results of this study, I recommend this study to be published so that academic researchers, authors, reviewers and most importantly the students.

Reading this study it advances current knowledge.

English Level: The English language is appropriate and understandable

We sincerely appreciate the reviewer’s positive overall evaluation and their recommendation for publication. It is highly rewarding to know that our work is considered relevant not only for researchers, authors, and reviewers, but also for students, who represent the next generation of professionals in the field. We are pleased that the language and clarity of the manuscript meet the expected standards, and we remain committed to producing research that advances current knowledge.

We trust that the improvements made based on the valuable feedback received will further enhance the quality and impact of this study, and we look forward to its dissemination within the scientific community.

Once again, thank you very much for the time spent and the interest shown in this work; as well as in the positive evaluations you have given of it.

Receive a warm greeting,

The authors.

Reviewer 4 Report

Comments and Suggestions for Authors

Title of the paper: Effect of COVID-19 on falls in a residential care facility for the elderly: Longitudinal observational study

Journal: Journal of Clinical Medicin

Authors: Ana González-Castro, Raquel Leirós-Rodríguez, Marta Nistal-Martínez, Ernesto Bodero-Vidal, Pablo Hernandez-Lucas

Review:

The manuscript lacks a clear and well-developed theoretical framework. While the introduction provides background statistics and contextual information, it does not establish a robust conceptual basis or integrate relevant theoretical models that would guide the study’s design and interpretation.

The methodology section is superficial and lacks sufficient detail. Key elements such as sampling rationale, data collection procedures, and analytical justification are presented in a vague manner, which undermines reproducibility and scientific rigour. The statistical analysis is minimal and lacks an in-depth explanation of variable selection or control of potential confounders.

The scientific contribution of the manuscript is very limited. The results are descriptive and do not provide novel insights beyond what is already known in the literature. The discussion section essentially reiterates known facts without offering substantial critical interpretation or theoretical advancement.

Author Response

Dear Editor and Reviewers of the Journal of Clinical Medicine:

Thank you very much for your suggestions and contributions to improving the quality of the manuscript. Following your indications, we respond, point by point, to the reviewers' comments.

In the text, all the modified or added sentences have been written in red to facilitate the correction by the reviewers.

Reviewer 4:

  1. The manuscript lacks a clear and well-developed theoretical framework. While the introduction provides background statistics and contextual information, it does not establish a robust conceptual basis or integrate relevant theoretical models that would guide the study’s design and interpretation.

We greatly appreciate your thoughtful observation regarding the need for a clearer and more developed theoretical framework. Following your valuable suggestion, we have substantially strengthened the Introduction section by incorporating an explicit conceptual foundation based on the biopsychosocial model. This framework integrates biological, psychological, social, and environmental determinants of fall risk in institutionalized older adults, supported by recent literature (2020–2025).

In particular, we have elaborated on how intrinsic factors (e.g., physical frailty, balance impairments, comorbidities), psychosocial aspects (e.g., social isolation, disruptions to daily routines), and environmental or organizational characteristics of residential facilities interact to influence fall risk. We have also emphasized how the COVID-19 pandemic simultaneously altered these components—limiting mobility, modifying routines, affecting supervision, and increasing loneliness—thereby highlighting the relevance of this multifactorial approach to understanding our study’s findings.

We believe these additions provide a more robust theoretical basis for the study design, contextualize the interpretation of the results, and reinforce the importance of our research in guiding targeted fall prevention strategies and preparedness for future health emergencies.

  1. The methodology section is superficial and lacks sufficient detail. Key elements such as sampling rationale, data collection procedures, and analytical justification are presented in a vague manner, which undermines reproducibility and scientific rigour. The statistical analysis is minimal and lacks an in-depth explanation of variable selection or control of potential confounders.

We are truly grateful for this insightful observation, which has helped us strengthen the transparency, reproducibility, and scientific rigor of our study. In response to your valuable feedback, we have expanded the Methods section to provide more precise details on the sampling rationale, data collection procedures, and analytical justification.

Specifically, we have:

  • Clarified the inclusion and exclusion criteria to enhance methodological transparency.

  • Provided a more explicit description of how fall-related variables were recorded and documented within the facility.

  • Added further detail on the statistical approach, including the rationale for selecting non-parametric tests and the Wilcoxon signed-rank test for paired comparisons.
  • Indicated that potential confounding factors were reviewed descriptively through year-by-year and contextual pattern analysis, noting that no multivariate adjustment was applied due to the study’s scope and sample size.

These additions aim to ensure that our methodology can be more easily reproduced and to better justify our analytical choices.

  1. The scientific contribution of the manuscript is very limited. The results are descriptive and do not provide novel insights beyond what is already known in literature. The discussion section essentially reiterates known facts without offering substantial critical interpretation or theoretical advancement.

We sincerely thank the reviewer for this insightful and constructive observation, which we fully acknowledge as an opportunity to strengthen the scientific value of our work. We agree that the initial version of the Discussion could have provided a deeper critical interpretation and a clearer theoretical contribution beyond the descriptive presentation of results.

Thanks to this valuable feedback, we have substantially expanded the Discussion to integrate our findings within the broader international literature, highlight possible alternative explanations for the observed trends, and explore their theoretical and practical implications for understanding fall patterns during periods of mobility restriction. We have also drawn connections to recent studies (2022–2024) conducted in similar contexts, with the aim of offering a more robust conceptual framework.

We are deeply grateful for the reviewer’s thoughtful assessment, which has been instrumental in guiding these improvements. Your comments have not only enhanced the quality and interpretative depth of our work but have also reinforced its relevance for clinical practice and public health. We truly appreciate the time, expertise, and dedication invested in reviewing our manuscript, and we remain sincerely thankful for the opportunity to improve it through your guidance.

Once again, thank you very much for the time spent and the interest shown in this work; as well as in the positive evaluations you have given of it.

Receive a warm greeting,

The authors.

Round 2

Reviewer 1 Report

Comments and Suggestions for Authors
  1. the discussion mentions an increase in falls while residents were accompanied, but it remains unclear who was present (a caregiver, another resident, a visitor?)
  2. The increase in nighttime falls is highlighted but not fully explored (Could reduced nighttime staffing or altered supervision protocols have contributed?Did sleep disturbances or changes in residents’ routines affect fall timing?Was lighting in private areas assessed during the study?)
  3. Add a short subsection titled “Clinical Implications” or “Policy Recommendations”

Author Response

Dear Editor and Reviewers of the Journal of Clinical Medicine:

Thank you very much for your suggestions and contributions to improving the quality of the manuscript. Following your indications, we respond, point by point, to the reviewers' comments.

In the text, all the modified or added sentences have been written in red to facilitate the correction by the reviewers.

Reviewer 1:

  1. The discussion mentions an increase in falls while residents were accompanied, but it remains unclear who was present (a caregiver, another resident, a visitor?).

We sincerely thank you for your thoughtful observation, which enabled us to identify a potential ambiguity in the definition of the variable “accompaniment.” We have now clarified in the Methods section that this term refers exclusively to the presence of staff members or external visitors at the time of the fall, and not to other residents. In addition, we have expanded the Discussion to emphasize that, although supervision has been described as an important factor in reducing adverse events in nursing homes, it does not necessarily guarantee the prevention of falls. To support this interpretation, we have incorporated an additional reference and highlighted the need for future research to examine how the type and quality of accompaniment may influence fall risk. We believe these revisions enhance both the clarity and the overall contribution of the manuscript.

Changes in the manuscript: Materials and Methods (lines 119–120); Discussion (lines 225–237).

  1. The increase in nighttime falls is highlighted but not fully explored (Could reduced nighttime staffing or altered supervision protocols have contributed?Did sleep disturbances or changes in residents’ routines affect fall timing?Was lighting in private areas assessed during the study?)

We sincerely thank you for this thoughtful comment, which enabled us to refine the interpretation of our findings. We have clarified in the Methods section that, according to institutional reports, nighttime staffing levels and supervision protocols remained unchanged between the pre-pandemic and pandemic periods. Therefore, the observed increase in nocturnal falls cannot be attributed to variations in supervision. However, we acknowledge that other factors not directly measured in this study—such as sleep disturbances, disrupted routines, and lighting conditions in private rooms—may have contributed. To address this, we have expanded the Discussion to incorporate these potential explanations, supported by recent literature linking poor sleep quality and altered routines to increased fall risk in older adults, as well as evidence demonstrating that adaptive lighting systems in nursing homes can substantially reduce nighttime falls. We believe these additions strengthen the contextualization of our results and underscore important directions for future research.

Changes in the manuscript: Materials and Methods (lines 123–124); Discussion (lines 255–260).

  1. Add a short subsection titled “Clinical Implications” or “Policy Recommendations”.

We sincerely thank you for this valuable suggestion, which we believe has significantly strengthened the manuscript. In response, we have added a new subsection entitled “Clinical Implications” at the end of the Discussion. This subsection summarizes the practical relevance of our findings, emphasizing the importance of implementing structured physical activity programs during confinement periods, enhancing nighttime supervision with adaptive lighting, reviewing environmental risk factors in private rooms, and integrating psychosocial support into fall-prevention strategies. From a policy perspective, it also highlights the need to embed fall-prevention protocols into contingency plans for future public health emergencies. We are grateful for this recommendation, which has enhanced the applied and translational value of our work.

Changes in the manuscript: Discussion (lines 297–305).

Once again, thank you very much for the time spent and the interest shown in this work; as well as in the positive evaluations you have given of it.

Receive a warm greeting,

The authors.

Reviewer 4 Report

Comments and Suggestions for Authors

Dear Editors,
I believe that the authors have significantly improved the manuscript compared to the previous version. Although the study remains methodologically and statistically limited, if the other reviewers agree and the editorial board considers the manuscript to meet the required standards. All major shortcomings that could be addressed have been corrected by the authors.

Author Response

Dear Editor and Reviewers of the Journal of Clinical Medicine:

Thank you very much for your suggestions and contributions to improving the quality of the manuscript. Following your indications, we respond, point by point, to the reviewers' comments.

In the text, all the modified or added sentences have been written in red to facilitate the correction by the reviewers.

Reviewer 4:

  1. I believe that the authors have significantly improved the manuscript compared to the previous version. Although the study remains methodologically and statistically limited, if the other reviewers agree and the editorial board considers the manuscript to meet the required standards. All major shortcomings that could be addressed have been corrected by the authors.

We sincerely thank you for your kind words and for recognizing the efforts we have made to revise and improve the manuscript. We greatly appreciate the time you devoted to the critical evaluation of our work and the clarity of your observations, which enabled us to strengthen the methodological and statistical aspects as much as possible within the constraints of the original design. We are pleased to know that the modifications introduced have met your expectations and that you consider the major shortcomings that could be addressed to have been satisfactorily corrected.

Once again, thank you very much for the time spent and the interest shown in this work; as well as in the positive evaluations you have given of it.

Receive a warm greeting,

The authors.